# Polymeric Carbon Nitrides for Photoelectrochemical Applications: Ring Opening-Induced Degradation

**DOI:** 10.3390/nano13071248

**Published:** 2023-03-31

**Authors:** Florentina Iuliana Maxim, Eugenia Tanasa, Bogdan Mitrea, Cornelia Diac, Tomáš Skála, Liviu Cristian Tanase, Cătălin Ianăși, Adrian Ciocanea, Stefan Antohe, Eugeniu Vasile, Eugenia Fagadar-Cosma, Serban N. Stamatin

**Affiliations:** 13Nano-SAE Research Centre, University of Bucharest, Atomistilor 405, 077125 Magurele, Romania; 2Department of Oxide Materials and Nanomaterials, Faculty of Applied Chemistry and Material Science, University POLITEHNICA of Bucharest, 060042 Bucharest, Romania; 3Department of Surface and Plasma Science, Charles University, V Holešovičkách 2, 18000 Prague, Czech Republic; 4National Institute of Materials Physics, Atomistilor 405A, 077125 Magurele, Romania; 5“Coriolan Drăgulescu” Institute of Chemistry, Mihai Viteazul Ave. 24, 300223 Timisoara, Romania; 6Hydraulics and Environmental Engineering Department, University POLITEHNICA of Bucharest, 060042 Bucharest, Romania; 7Faculty of Physics, University of Bucharest, Atomistilor 405, 077125 Magurele, Romania; 8Academy of Romanian Scientists (AOSR), Ilfov No 3, 050094 Bucharest, Romania

**Keywords:** energy, photo-electrochemistry, degradation, carbon nitrides

## Abstract

Active and stable materials that utilize solar radiation for promoting different reactions are critical for emerging technologies. Two of the most common polymeric carbon nitrides were prepared by the thermal polycondensation of melamine. The scope of this work is to investigate possible structural degradation before and after photoelectrochemical testing. The materials were characterized using synchrotron radiation and lab-based techniques, and subsequently degraded photoelectrochemically, followed by post-mortem analysis. Post-mortem investigations reveal: (1) carbon atoms bonded to three nitrogen atoms change into carbon atoms bonded to two nitrogen atoms and (2) the presence of methylene terminals in post-mortem materials. The study concludes that polymeric carbon nitrides are susceptible to photoelectrochemical degradation via ring opening.

## 1. Introduction

Sun-driven electrochemical reactions, such as: (1) photocatalytic environmental remediation [1], (2) photocatalytic synthesis [2], (3) photoelectrochemical water splitting to generate hydrogen fuel [3,4] and (3) photoelectrochemical CO_2_ reduction to produce chemical feedstocks [5,6], will have the utmost importance in securing sustainability. The electron-hole pairs generated by illuminated semiconductors are the key to unlocking the full potential of photoelectrochemistry. More than 50 years have passed since the seminal paper by Fujishima and Honda [7] that led to intense research in finding materials with high photoelectrochemical activity, such as La-doped NaTaO3 [8]. Further on, the attention was focused on finding cheap and abundant materials such as polymeric carbon nitride [9] which has been in the spotlight for more than a decade.

Polymeric carbon nitrides are obtained from the thermal polycondensation of nitrogen-rich precursors, such as melamine [10], cyanamide [11] and other similar compounds, such as 2,4,6-triazido-1,3,5-triazine [12]. The structure and surface chemistry of carbon nitrides obtained by thermal polycondensation has been the subject of intense debate in the literature [13,14,15,16,17,18,19]. It was initially believed that the product of the polycondensation is the highly stable carbon nitride (i.e., C_3_N_4_). Ideal C_3_N_4_ consists of a carbon–nitrogen network which is challenging to obtain by heat treatment due to the presence of hydrogen and other impurities in the precursors and air. Indeed, it was shown that C_3_N_4_ contains hydrogen below 3% and its structure resembles that of melon (i.e., C_6_N_9_H_3_) [18,19]. The term graphitic carbon nitride (i.e., g-C_3_N_4_) is still used in the literature, albeit the structure and surface chemistry are fundamentally different. Carbon nitride is inherently more stable than melon, which made stability issues improbable, and, therefore, such studies are not present in the literature. The term g-CN will be used instead of g-C_3_N_4_ in the work at hand to underline the fundamental difference between the two classes of materials.

The photoelectrochemical activity of pristine carbon nitride is not that high but can be further tuned by doping, annealing, increased surface area, chemical modifications, etc. [3]. The nature of active sites and fundamental reaction mechanisms are very scarce in the literature [4], despite the large body of work available in the literature on polymeric carbon nitrides. Stability investigations are paramount in establishing active sites and reaction mechanisms, as they can bring to light reactant species. It was shown that the nitrogen concentration decreases during photocatalytic experiments which indicates that polymeric carbon nitrides are not stable under photocatalytic conditions, that is, powder dispersed in a liquid [20]. In contrast to photocatalysis, photoelectrocatalytic conditions mean that a controlled potential is applied to a photoelectrode. Polymeric carbon nitride photoelectrodes are obtained by spray deposition on a transparent fluorine-doped tin oxide (FTO) glass. It is obvious that at large potentials the interface between the carbon nitride layer and FTO will be severely affected, especially in gas-evolving experiments (e.g., hydrogen evolution reaction or CO_2_ reduction).

Polymeric carbon nitrides were initially tested for hydrogen evolution [9,21], a reaction that does not involve complex surface intermediates. Recently, the interest has shifted towards more complex reactions, such as CO_2_ reduction [6,22]. Electrochemical CO_2_ reduction is a considerably more complex reaction that involves many steps and adsorbed intermediates even on metallic surfaces [23,24]. Electrochemical reduction reactions at neutral pH usually start with the formation of radicals, such as superoxide radicals or CO_2_ anion radicals [24,25]. It is expected for a defective carbon-based structure to be prone to degradation that can alter the initial morphology and surface chemistry which leads to deactivation. In this respect, carbon nitride stability investigations are vital in reaching beyond the state of the art.

Herein, two common polymeric carbon nitrides, crystalline and amorphous (inset Figure 1C,D), were prepared by the thermal polycondensation route. Subsequent annealing of the polymeric carbon nitride resulted in what is known as red or amorphous carbon nitride [26,27]. Despite having a similar structure, characterization by synchrotron radiation showed differences in surface chemistry while electron microscopy highlighted different morphologies. We made use of the differences in similarities to understand degradation mechanisms, a phenomenon that is currently not receiving much interest. Post-mortem FT-IR revealed structural deformation. The study at hand shows that: (1) annealing eliminates unstable triazines and heptazine, (2) carbon–nitrogen chemistry is irreversibly changed after photoelectrochemical degradation and (3) methylene terminals are present in the degraded samples. The latter conclusion suggests that nitrogen concentration is smaller in photoelectrochemically degraded samples. We conclude that ring-opening causes the distortion of the nitrogen local structure leading to changes in morphologies.

## 2. Results

### 2.1. Material Synthesis

Elemental analysis showed that graphitic carbon nitride (g-CN) consisted of 35.5% carbon, 62.6% nitrogen and 1.1% hydrogen. The as-obtained g-CN was placed in a horizontal tube furnace at 675 °C with a constant Ar flow, hereinafter g-CN-HT (Figure 1). Elemental analysis showed a slightly different C/N/H ratio for g-CN-HT, that is, 35.8% carbon, 62.0% nitrogen and 0.7% hydrogen. The carbon-to-nitrogen ratio (C/N) was determined to be 0.57 and 0.58 for g-CN and g-CN-HT, respectively. The C/N theoretical values for melon and g-C_3_N_4_ are 0.67 and 0.75, respectively, significantly higher than the samples under investigation. Elemental analysis showed that the samples are similar in bulk composition, although considerably different in structure, morphology and surface chemistry (*vide infra*).

### 2.2. Structure and Morphology

The structure of polymeric carbon nitrides can be qualitatively assessed by Fourier transform–infrared (FT-IR) [13,28,29,30], shown in Figure 1A. Heptazine (i.e., cyameluric nucleus) can be identified in the FT-IR by its specific absorption band at 810 cm^−1^ which was visible for both samples. The region between 1200 and 1700 cm^−1^ has six main peaks corresponding to different bonds in the heterocycle. The peaks at 1233 cm^−1^ and can be attributed to the nitrogen bonded to three carbons in the trigonal units, as in C-N(-C)-C. The bands at 1457 and 1571 cm^−1^ correspond to the ring vibrations. The peaks at 1403 and 1637 cm^−1^ correspond to the δ(NH) and δ(NH_2_), respectively. The last region defined between 2800 and 3400 cm^−1^ can be assigned to ν(NH).

X-ray diffraction (XRD) was used to probe the long-range order of the materials under investigation (Figure 1B). Four peaks were observed in the region between 2θ = 12° and 2θ = 32°. The peak at 2θ = 27.76° has the highest intensity followed by the peak at 2θ = 13.12°. Previous XRD studies showed that the polymeric carbon nitride resulted from solid-state pyrolysis of melamine is in fact melon, although the presence of heptazine-based g-C_3_N_4_ cannot be fully excluded on the sole basis of XRD [14,18,19]. The XRD profile of g-CN indicated an orthorhombic geometry with main diffractions at 2θ = 27.76° and 2θ = 13.12° which were assigned to (002) and (210), respectively. Available structural models showed that the separation between parallel melem units gives rise to (210) while (002) is attributed to the separation between the graphitic sheets [14,19]. An interlayer distance of 3.22 Å was obtained for g-CN based on the (002) Bragg peak which is similar to other reported values [18,19]. The interlayer distance for g-CN-HT was determined to be 3.19 Å, resulting from the 0.16° increase in 2θ (inset of Figure 1B). The intensity ratio between (210) and (002), I_210_/I_002_, is another important feature in the XRD profiles of carbon nitrides [14]. The calculated I_210_/I_002_ was 0.17 and 0.20 for g-CN and g-CN-HT, respectively. The I_210_/I_002_ values are similar to the theoretical value, that is, 0.19.

The morphology of g-CN and g-CN-HT (Figure 1C,D, Appendix A) was assessed by electron microscopy. SEM and TEM investigation exposed the tube-like morphology of g-CN (Appendix A and black arrows in Appendix A). Representative SEM images of g-CN showed that tubular structures were bundled into larger formations (Appendix A). A fractured piece showed cracked tubes that were well-aligned and tightly packed together (Appendix A). The length of the tubes was determined to be around 2–5 μm with a tube diameter below 300 nm (Appendix A). The tubes were densely packed into ordered structures mixed with an amorphous component. Individual tubes were observed only in the proximity of an amorphous fraction (black arrows in Appendix A).

The morphology of g-CN-HT is composed of thin sheets lumped into larger fractions (Appendix A). TEM investigations of g-CN-HT (Figure 1D) showed an amorphous structure without any order which is further supported by the SEM (Appendix A). Further TEM investigations revealed that the tubes present in g-CN have completely lost their tubular geometry (red arrows in Appendix A). Similar morphological features were encountered for amorphous and red carbon nitrides [25,26].

The textural parameters determined using nitrogen (N_2_) adsorption–desorption isotherms (Appendix A) indicate, considering IUPAC recommendations [31,32], a type IVa isotherm with an H3 hysteresis loop. Inflection points in such hysteresis are not obvious. Nearly parallel adsorption and desorption curves are expected for materials with plate-like particles, namely: tapered plate pores with both opening ends coexisting with narrow parallel-plate pores [33]. Both samples belong to mesoporous materials. A BET surface area of 20 m^2^/g and 47 m^2^/g was determined for g-CN and g-CN-HT, respectively. A pore volume of 0.09 mL/g and 0.12 mL/g was measured for g-CN and g-CN-HT, respectively. The pore size distribution obtained from the BJH method (see Appendix A) showed a slight increase from 3.92 nm to 3.95 nm for g-CN and g-CN-HT, respectively. The textural parameters indicate the formation of new and bigger pores in agreement with the results obtained by TEM and SEM analysis.

### 2.3. Surface Chemistry Characterization by Synchrotron Radiation

Synchrotron-based near-edge X-ray absorption fine structure (NEXAFS) was used to characterize the chemistry of the materials under investigation (Figure 2). The reader should bear in mind that NEXAFS on carbon nitrides is still a topic of hot debate in the literature [30,34,35,36]. NEXAFS was used in this work as a fingerprinting technique to map the nitrogen chemistry to deconvolute the N 1s XPS (*vide-infra*). The NEXAFS for g-CN and g-CN-HT showed a similar profile to previous reports [30,34,35,36]. The highest intensity peak at 399.3 eV (N_1_) is followed by two smaller features around 401 eV (N_2_ and N_3_) and the second most intense peak at 402.1 eV (N_4_). The largest contribution to the NEXAFS (N_1_) was ascribed to the sp^2^ hybridized nitrogen, as in C=N-C in the heptazine unit [35], which is also the most abundant nitrogen bonding in the structure (vide-infra). Peak N_2_ was assigned to the central tertiary nitrogen, as in (C)_3_-N in heptazine. Peak N_3_ was assigned to terminal nitrogen as in C-NH_2_ [35]. It is generally accepted that the nitrogen bridging heptazine units, as in (C)_3_-N, have a resonance of around 402 eV [30]; however, it was shown recently that it overlaps with the sp^2^ hybridized nitrogen [35,36]. In consequence, peak N_4_ was assigned to the sp^2^ hybridized nitrogen [35]. Peak assignment in NEXAFS (Figure 2) was consistent with the most recent equivalent core-hole time-dependent density functional theory modeling [35] which showed the contribution of four different nitrogen chemistry.

FT-IR and XRD showed that g-CN and g-CN-HT possess a similar structure while the SEM and TEM showed that their morphology differs significantly. Core-level X-ray photoelectron spectroscopy (XPS) was used to probe the surface chemistry of g-CN and g-CN-HT to achieve an improved understanding of the materials (Figure 3). Wide scan XPS showed that the samples contained only C, N, O and Si (Appendix A). Samples prepared for XPS were deposited on Si which explains the presence of Si in the wide scan.

The N 1s core-level spectrum of g-CN is presented in Figure 3A. The deconvolution of the N 1s XPS into four main contributions was in line with the NEXAFS findings and previous literature reports [18]: (1) the sp^2^ nitrogen located at 399.0 ± 0.1 eV; (2) the nitrogen in the amino group, as in C-NH_2_, located at 399.5 ± 0.1 eV; (3) the nitrogen linking two melem units, hereinafter as (C)_2_-NH, located at 400.5 ± 0.1 eV and (4) the central nitrogen bonded to three carbons in the central heptazine ring, hereinafter (C)_3_-N, located at 401.5 ± 0.1 eV. The component at a binding energy larger than 403.5 eV was attributed to a shake-off type of change in Coulombic potential, generally known as XPS satellite peaks. The area ratio normalized to the total intensity is presented in Table 1 for every peak. The previous elemental analysis and XRD investigations suggested that melon is a more representative structure for g-CN and g-CN-HT, rather than g-C_3_N_4_. Melon’s ideal structure consists of C=N-C, C-NH_2_, (C)_2_-NH and (C)_3_-N in 67:11:11:11 proportions [18] which was found to be very similar to g-CN, that is, 69.5:9.6:11.1:9.8 (Table 1). This underlines g-CN’s similarity to melon rather than g-C_3_N_4_, although there were minor deviations from the ideal structure of melon. A similar approach was pursued for the surface chemistry of g-CN-HT which was determined to be: 66.2:12.2:13.1:8.5 (Table 1). A decrease in the surface concentration of C=N-C and (C)_3_-N was observed between g-CN and g-CN-HT (Table 1) which corresponds to the breakage of the s-triazine and heptazine units. The increase in the surface concentration of (C)_2_-NH observed for g-CN-HT (Table 1) gave more substance to the disruption of the heptazine-based g-C_3_N_4_ structure. A similar rationale was used for the increase in the surface concentration of C-NH_2_ which indicates that the s-triazine units underwent a structural change. 

The focus was then turned to carbon’s core-level XPS spectra (Figure 3B,D). NEXAFS was used for fingerprinting which showed a main peak at 287.9 eV (C2) with a smaller feature at 287 eV (C1; Figure 2). The main peak, C2, was assigned to the absorbing carbon atom in N=C-(N)_2_ while distortions of the local structure were found to lead to a feature at smaller binding energy [35]. The XPS core-level spectra of C 1s for g-CN and g-CN-HT (Figure 3B,D) showed a main peak at 288.6 eV which was assigned to sp^2^ carbons of the heptazine rings, as in N=C-(N)_2_ as it was previously shown [18]. The second most intense peak at approx. 285 eV was assigned to adventitious carbon, as in C-C. Hydrocarbon impurities easily adsorb on the sample support or even on the sample which is usually associated with adventitious carbon. The peak labeled as impurities in Figure 3B,D was located at approx. 286–287 eV which is an area specific for oxygen moieties on carbon surfaces that can be related to oxidized hydrocarbons [37].

The carbon-to-nitrogen ratio (C/N) on the surface was determined by dividing the N=C-(N)_2_ area in C 1s by the nitrogen area, excluding satellites (Table 1). An ideal g-C_3_N_4_ should have a 0.75 C/N ratio, whereas the same ratio for melon should be 0.67 [18,19]. Both samples under investigation showed a C/N ratio smaller than the ideal g-C_3_N_4_ even in the first few nm. The bulk C/N ratio in both samples was below 0.6 (vide supra). The results point to a chemical structure and composition similar to that of melon at the nanoscale, albeit significantly different in bulk.

### 2.4. Photoelectrochemical Characterization

The UV-Vis spectra (Figure 4A) exhibited the typical carbon nitride profile with a semiconductor bandgap onset. g-CN showed a main peak at 370 nm consistent with the yellow color. The π-π* electron transition gives rise to the peak at 370 nm (Figure 4A). The absorption of g-CN-HT was at 520 nm which is also known as a red-shift. The n-π* electron transitions involving lone pairs on the N atoms situated at the edge are believed to be the source of the 520 nm peak [25,26,38]. The reader should bear in mind that n-π* electron transitions are forbidden in ideal structures based on s-triazine repeating units [39,40]. Tauc plots were constructed to obtain the optical bandgap (inset Figure 4A). The *x*-axis intercept of the linear region in the Tauc plot showed optical bandgaps of 2.60 eV and 2.21 eV for g-CN and g-CN-HT, respectively. Similar bandgap values were obtained for pristine and heat-treated carbon nitrides [38,41]. Carbon nitrides with a narrower bandgap have been obtained before by heat treatment or alkaline earth metal doping [25,26,38,40].

The flat band potential, V_fb_, can be obtained from the electrochemical impedance spectroscopy measured in the dark [42]. The specific capacitance is related to the electrode-applied potential by the Mott–Schottky equation (see Appendix A). The values reported in this work are shifted by −0.61 V (−0.2 V for the Ag/AgCl reference potential and −0.41 V for the potential pH correction) from the values reported against the normal hydrogen electrode (NHE) at pH = 0 [41]. Typical values for the electrochemical conduction band of carbon nitrides were found to span between −1 and −1.2 V vs. NHE which is equivalent to −1.61 and −1.81 V vs. Ag/AgCl [41] (taking into account the corrections mentioned above).

The photocurrent density is smaller than 0.6 µA cm^−2^ for both samples (Figure 4C). The maximum photocurrent density for g-CN and g-CN-HT was found at −0.4 V vs. Ag/AgCl and −0.7 V vs. Ag/AgCl, respectively. There are many factors at play in generating the photocurrent, such as the photoelectrode loading and the deposition method. Nevertheless, photocurrent densities ranging from 0.1 µA cm^−2^ to 10 µA cm^−2^ are common in the literature [38,43,44,45]. The trend in photocurrent density is steadily increasing with increasing potential for g-CN-HT. A decreasing trend in photocurrent density was observed for g-CN after reaching the maximum at −0.4 V vs. Ag/AgCl. Figure 4D shows the electronic band structure for g-CN and g-CN-HT based on the bandgap determined from the Tauc plot (inset Figure 4A) and the Mott–Schottky plot (Figure 4B). The right-hand axis in Figure 4D shows the most probable electrochemical reactions to take place at the photoelectrodes presented in Figure 4C. All the reactions have a standard potential between −0.3 and −0.8 V vs. Ag/AgCl. Other reactions were excluded to simplify the image considering that product formation and selectivity fall beyond the scope of this manuscript, that is, photoelectrochemical corrosion. The conversion efficiency and product selectivity during the photoelectrochemical CO_2_ reduction reaction over carbon nitrides has been extensively studied [38,41,45]. However, little is known about the photoelectrochemical stability of carbon nitrides.

### 2.5. Post-Mortem Analysis

Degradation mechanisms in carbon materials are mostly related to carbon corrosion, which can be traced to changes in chemistry and structure [46]. We showed that nitrogen and oxygen moieties should be considered as electrochemically active sites [37]. One of the characterization methods that can test simultaneously nitrogen and oxygen moieties is FT-IR [47]. The post-mortem structure of g-CN and g-CN-HT was assessed by FT-IR (Figure 5). The raw FT-IR data can be found in the Appendix A. Sodium bicarbonate could not be used as an electrolyte in the post-mortem experiments due to the presence of carbonate ions which have active IR bands between 500 and 2000 cm^−1^ (Appendix A). In this respect, sodium sulfate was used as the electrolyte which has two active bands visible in the post-mortem samples at approx. 600 and 1100 cm^−1^ (Figure 5 and Appendix A). Adsorbed water can be clearly observed at 3500 cm^−1^ only in the post-mortem samples (red curves in Figure 5A,B). Changes in structure can be qualitatively assessed by normalization to the largest feature, that is, the peak at 1233 cm^−1^ (with zero transmittance in Figure 5A,B) specific for the nitrogen bonded to three carbons in the trigonal units, as in C-N(-C)-C or bridging C-(NH)-C units [19].

It can be clearly observed that the peaks specific for heptazine (810 cm^−1^), amine (3000–3350 cm^−1^) and ring (1570 cm^−1^) vibrations have decreased in transmittance (Figure 5). To further quantify the changes in structure, the loss percentage was calculated in Figure 5C by normalizing the difference in the absorbance to the initial absorbance. A loss in amines between 45 and 50% was determined for both samples. The intensity at 810 cm^−1^, that is, heptazine-specific, decreased by 21 and 7% for g-CN and g-CN-HT, respectively. The intensity for ring vibrations (1570 cm^−1^) decreased by almost 17% and 4% for g-CN and g-CN-HT, respectively (Figure 5C).

A closer investigation of the region between 2000 and 4000 cm^−1^ unraveled additional structural changes (Appendix A). The intensity of the peak at approx. 3450 cm^−1^ increased considerably for both samples which was ascribed to OH. Most probably, there is a mix of the OH sources: (1) adsorbed water on the surface and (2) the OH part of the structure. It follows suit that the water-based electrolytes will adsorb on the surface of the post-mortem samples which is confirmed by the presence of the sodium sulfate-specific peaks at 600 and 1100 cm^−1^ (Figure 5). The CO_2_-specific doublet appears at approx. 2450 cm^−1^ which was ascribed to the CO_2_ gas present in the FT-IR testing chamber. Another doublet appeared at 2853/2922 cm^−1^ and 2852/2922 cm^−1^ for post-mortem g-CN and g-CN-HT, respectively, which is specific for ν(-CH_2_) (Appendix A). The reader should bear in mind that carbon-hydrogen bonds are forbidden in the heptazine-based carbon nitride and the melon theoretical model. The methylene groups were not detected in the initial samples which is consistent throughout the literature [3,10].

The peak centered at 1206 cm^−1^ increased in transmittance (Appendix A) which means an increase in the concentration of this specific carbon moiety after photoelectrochemical experiments. The C-N stretching in amines or C-O stretching can have vibrations in this region. Considering that the amine intensity (located at 300–3350 cm^−1^) is decreasing, the peak at 1206 cm^−1^ can then be ascribed to C-O. In fact, it was expected for minor oxygen moieties to be present in the initial structure due to the synthesis that took place in a covered crucible and, therefore, exposed to the oxygen partial pressure in air. The peak at 1206 cm^−1^, now assigned to C-O, is at the beginning of the region specific for the triazine heterocycle (1200–1700 cm^−1^) which makes the quantification dependent on baseline subtraction. To avoid misguiding the reader, we will look at this peak only qualitatively, that is, an increase in the C-O concentration irrespective of the sample following the photoelectrochemical experiments. The last mentionable difference between the initial and post-mortem samples is the region between 950 and 1050 cm^−1^ (Appendix A). The appearance of the peak at 990–995 cm^−1^ was assigned to the -N-C-N-, herein N_2_C, based on the most recent literature report [48].

## 3. Discussion

We showed, based on the photoelectron spectroscopy and FT-IR, that g-CN and g-CN-HT have a distorted structure. The materials cannot be classified as g-C_3_N_4_ because they do not possess only carbon–nitrogen bonds; hydrogen and oxygen are also in the structure (see Section 3). Moreover, the C:N ratio of g-CN and g-CN-HT is below 0.6 which is in stark contrast to the theoretical value of g-C_3_N_4_ (i.e., 0:75). Both samples have a structure similar to that of melon, albeit with different morphologies and surface area as indicated by the electron microscopy and BET analysis, respectively. The nature of the C-O bonds in the initial structure (i.e., part of the edge or the ring) remains elusive and falls beyond the scope of this work.

Post-mortem FT-IR confirmed that the changes in morphology arise from the structure of the initial materials. The transmittance of the most stable nitrogen moiety, that is, the trigonal nitrogen bonded to three carbons at the heptazine’s core, does not change upon photoelectrochemical experiments, which validates the post-mortem FT-IR. The peak from 3000 to 3350 cm^−1^ brings at least two amines in discussion: (1) primary amines at the edge of the polymeric network, that is, N=C(-NH_2_)-N, and (2) secondary amines bridging C-(NH)-C units. When at least two amines are lost per heptazine unit, the heptazine will partly lose its C=N bonds, and it will force one of the rings to open which will result in a methylene terminal. Such a mechanism is consistent with the loss of the triazine ring and heptazines (Figure 5C) and with the presence of methylene peaks in the post-mortem samples (Appendix A). Moreover, the peak at 990–995 cm^−1^ (Appendix A), previously assigned to N_2_C, supports the ring opening mechanism. An XPS study on the stability of polymeric carbon nitrides showed that the nitrogen concentration decreased while the oxygen concentration increased [20]. Such findings were confirmed for both samples in Appendix A. The location of the oxygen atoms in the initial and post-mortem structures is still elusive.

Which material is more stable? Figure 5 clearly shows that both materials are prone to corrosion. The degradation of g-CN can be rationalized by the loss of the heptazine structure and amines (vide supra) but this mechanism is not valid for g-CN-HT, which is intriguing. The larger amine loss without the distortion of the triazine and heptazine (Figure 5C) in g-CN-HT can be traced to the difference in surface chemistry between the two samples (see Section 3). The concentration decrease in C=N-C and (C)_3_-N (Table 1) is an indication that part of the triazine and heptazine was transformed upon annealing which is confirmed by the concentration increase in C-NH_2_ and (C)_2_-NH (Table 1). The remaining trigonal nitrogen, (C)_3_-N, is stable up to 675 °C. Therefore, the annealing procedure can be viewed as a “filtration” of the unstable trigonal nitrogen moieties. Moreover, the full-width half maximum of C 1s is visibly larger for g-CN-HT than for g-CN (Figure 3B,D). Only photoelectrons extracted from carbon atoms with different binding energy in the N=C-(N)_2_ moiety can explain the increase in the full-width half maximum, which points towards the deformation of the N=C-(N)_2_ local structure. The heat treatment (that the g-CN-HT was subjected to) caused defects in the initial structure and made amines bond to defective structures [26] and, therefore, more prone to degradation. Such mechanisms can explain the larger loss in amines and the smaller loss in the triazine ring and heptazine structure. To answer the question at the beginning of this paragraph, g-CN-HT preserves its initial backbone structure better than g-CN, albeit deforming the edge structure. The reader should bear in mind that g-CN-HT’s initial structure is significantly different from that of g-CN.

Post-mortem investigations carried out in this work showed that carbon nitride undergoes morphological and structural changes during photoelectrochemical reduction. It is well-known that the superoxide radical is the first electron transfer in an oxygen reduction reaction at nitrogenated carbon electrodes [49,50]. Similarly, CO_2_ needs to be activated through the one electron reaction to the CO_2_ anion radical [24]. The generation of hydroxyl radicals is expected as well at neutral pH under illuminating conditions. Oxygen traces in the electrolyte can lead to the generation of the superoxide radical. Triazine is considered the polymeric carbon nitride building block. Triazine is a well-known herbicide, and radical photolysis is one of the degradation approaches to reduce their environmental impacts [51,52]. It can be inferred that there are many radicals generated close to the electrode surface during photoelectrochemical experiments (i.e., −0.7 V vs. Ag/AgCl) which gives more substance to the photoelectrochemical degradation of carbon nitrides. The photoelectrochemical degradation mechanism is still elusive owing to several reactions taking place simultaneously, that is, oxygen, hydrogen, and carbon dioxide reduction reactions. In this respect, the carbon source of the product generated during (photo)electrochemical carbon dioxide reduction reaction should be carefully considered.

## 4. Conclusions

Two different polymeric carbon nitrides were synthesized by thermal polycondensation. A fundamental understanding of the structure and surface chemistry was achieved based on synchrotron radiation characterization, X-ray diffraction and advanced electron microscopy. The textural data reveal that by increasing the annealing temperature, the specific surface area is also increased. The electron band structure was determined by Tauc and Mott–Schottky plots. The product of thermal polycondensation is not the generally accepted graphitic carbon nitride or melon, although the product is structurally closer to the latter. The materials showed irregular photocurrent increase during the photoelectrochemical CO_2_ reduction experiments. This work delivers two main findings: (1) the carbon–nitrogen chemistry is irreversibly changed during photoelectrochemical applications and (2) methylene terminals were found in both samples after photoelectrochemical application. The presence of methylene terminals indicates that nitrogen can leave the carbon nitride structure. All in all, the work at hand shows that polymeric carbon nitrides are prone to degradation, albeit the degradation mechanism is not completely understood. The extensive use of polymeric carbon nitrides, especially doped carbon nitrides, in (photo)electrochemical CO_2_ reduction should take into account the carbon source of the resulting product.

## Figures and Tables

**Figure 1 nanomaterials-13-01248-f001:**
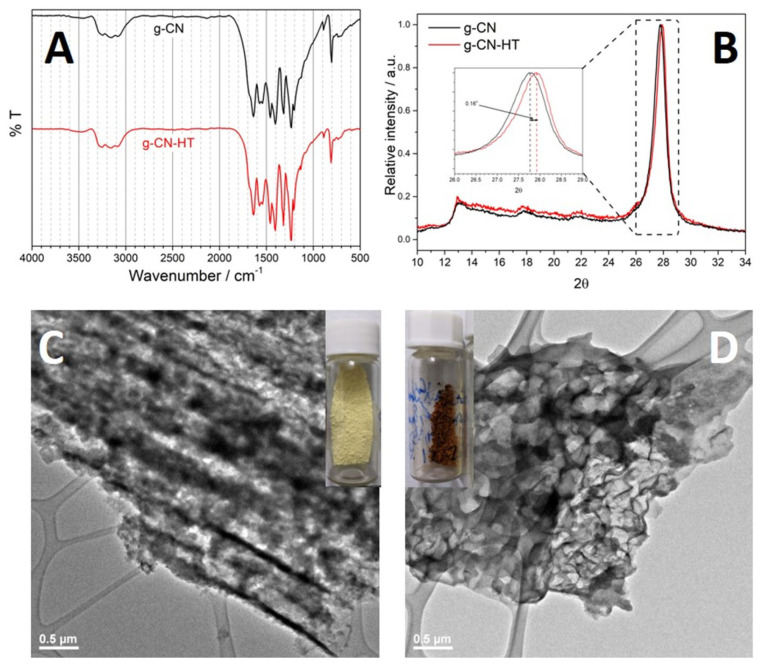
Structure and morphology of g-CN and g-CN-HT. Fourier-transform infrared spectroscopy of g-CN, black line, and g-CN-HT, red line (**A**); X-ray diffractograms of g-CN, black line, and g-CN-HT, red line (**B**); transmission electron microscopy of g-CN (**C**) and g-CN-HT (**D**).

**Figure 2 nanomaterials-13-01248-f002:**
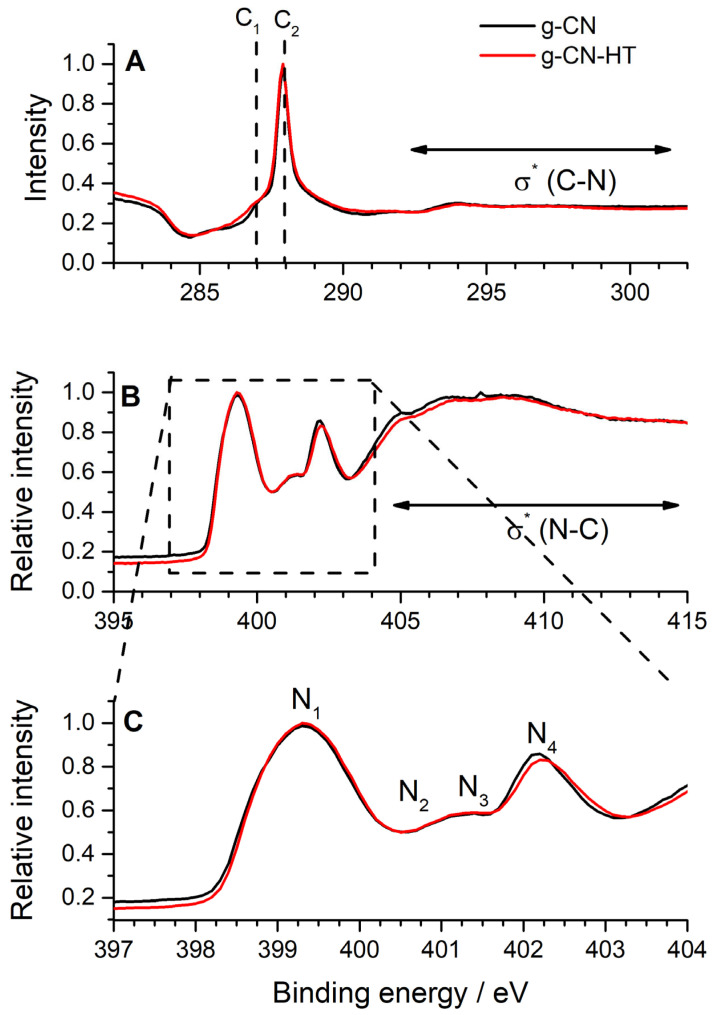
NEXAFS: C 1s K-edge; σ* at binding energies >292 eV refers to C1s→ σ* transitions (**A**), N 1s K-edge; σ* at binding energies larger than 405 eV refers to N1s→ σ* transitions (**B**) and detailed structure of the N 1s K-edge from 397 to 404 eV (**C**).

**Figure 3 nanomaterials-13-01248-f003:**
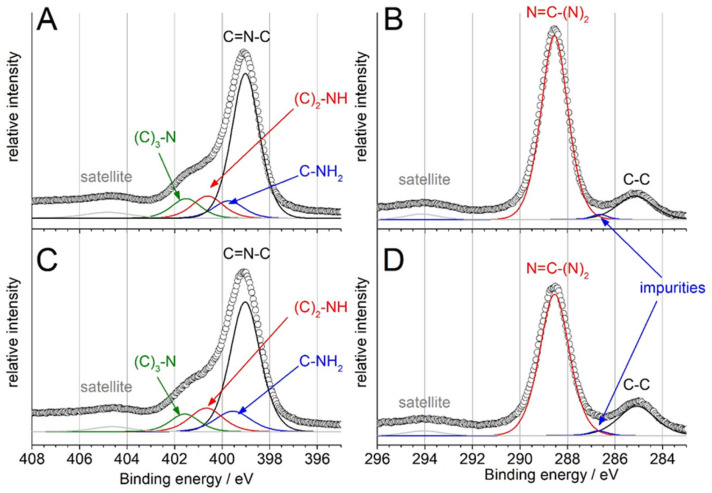
X-ray photoelectron spectroscopy (XPS) investigation on g-CN (**A**,**B**) and g-CN-HT (**C**,**D**). N 1s core-level spectra for g-CN (**A**) and g-CN-HT (**C**). C 1s core-level spectra for g-CN (**B**) and g-CN-HT (**D**).

**Figure 4 nanomaterials-13-01248-f004:**
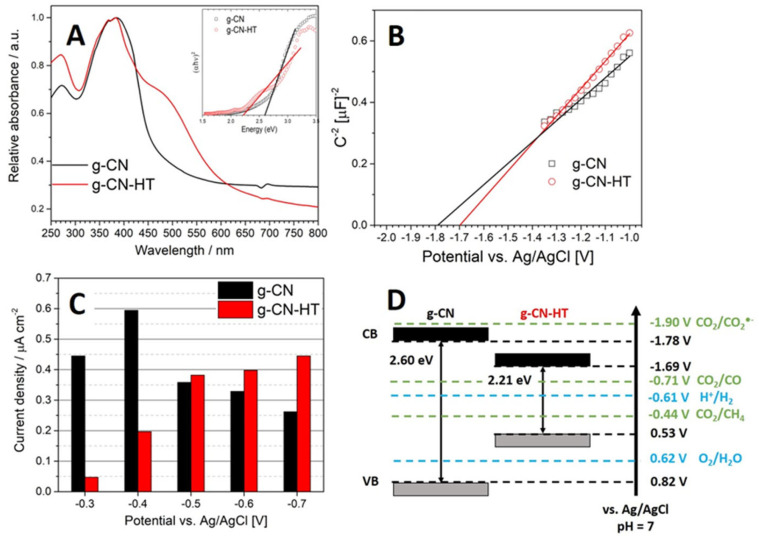
UV-Vis diffuse reflectance spectroscopy with the Tauc plot in the inset (**A**); Mott–Schottky plot obtained from dynamic electrochemical impedance spectroscopy at a 100 kHz frequency and 10 mV amplitude (**B**); photocurrent density determined from chronoamperometry experiments under illumination in CO_2_ saturated 0.5 M KHCO_3_ (**C**) and electronic band structure: grey band represents the valence band; black band represents the conduction band (**D**).

**Figure 5 nanomaterials-13-01248-f005:**
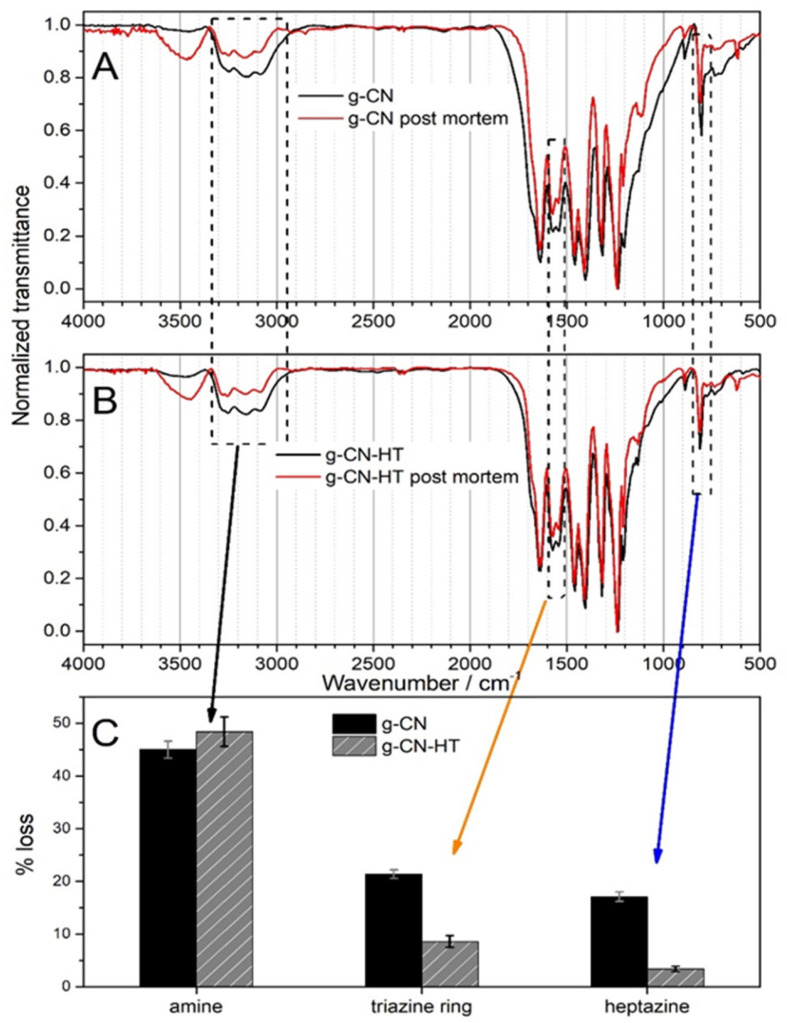
FT-IR investigations of g-CN (**A**) and g-CN-HT (**B**). The percentage losses of amine (black arrow), ring unit (orange arrow) and the triazine unit (blue arrow) are shown in (**C**) and were calculated as the difference in absorbance between the initial and post-mortem samples and divided by the absorbance of the initial sample.

**Table 1 nanomaterials-13-01248-t001:** Surface chemistry determined as% from the XPS core-level spectra of N 1s and C 1s.

	C/N	N 1s/%				C 1s/%		
		C=N-C	C-NH2	(C)2-NH	(C)3-N	C-C	imp	N=C-(N)2
g-CN	0.68	69.5	9.6	11.1	9.8	14.3	1.3	84.3
g-CN-HT	0.72	66.2	12.2	13.1	8.5	20.4	1.5	78.0

## Data Availability

The data presented in this study are available on request from the corresponding author.

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
