# Peer review of "Polymeric Carbon Nitrides for Photoelectrochemical Applications: Ring Opening-Induced Degradation"

_nanomaterials, 2023, doi:10.3390/nano13071248_

Round 1

Reviewer 1 Report

In this study, authors discuss "Polymeric carbon nitrides for photoelectrochemical applica-tions: ring opening induced degradation". Paper need some minor correction before final acceptance

1- Purity materials must be clear and detail of instruments must be add

2- Novelty work must be discuss in abstract and end paragraph of introduction section

3- Mechanism of system must be discuss

4- Selectivity of work must be test

5- conclusion must be improve with more data

Author Response

1- Purity materials must be clear and detail of instruments must be add

Purity has been added to every material. The instruments details have been completed. The entire Material and Methods section has been moved in the supplementary information.

2- Novelty work must be discuss in abstract and end paragraph of introduction section

The abstract and the introduction section has been improved according to the highlighted phrase in the revised manuscript.

3- Mechanism of system must be discuss

The degradation mechanism cannot be explained in detail at this moment without additional experiments. See next point.

4- Selectivity of work must be test

The manuscript at-hand is focused on the changes of the solid-state electrode. Indeed, carbon nitrides are degrading. It is possible that carbon and/or nitrogen fractions released in the electrolyte will influence the formed products. We can say at this stage that we determined the products by means of a GC coupled to FID and BDI. The measured products were not consistent. Preliminary results suggested that such materials might degrade. To deliver irrefutable results, additional experiments are needed. Detailed isotope labeling experiments are currently underway. We kindly rebut this comment at this stage.

5- conclusion must be improve with more data

The conclusion section has been improved. The changes are marked with yellow in the revised document.

Reviewer 2 Report

Dear authors, 

The manuscript is nicely written and the characterization of materials are well executed. These are few minor points which should be included before publication:
 - Figure 1 is named many times is the text, the first one in pg 2 but the figure is on pg 5., so it make tedious to have to search the figure everytime It might be more useful to separate this figure into two Figure 1 (a, b) and figure 2 (c, d).
- Figure S4 should be in the main manuscript. Moreover, in the text it is said "
The highest intensity peak at 399.3 eV (A) is followed by 2 smaller features around 401 eV (B and C)" but in the figure it could be not appreciate this two features. It could be useful to include a zoom of this regios as an inset.

Author Response

 - Figure 1 is named many times is the text, the first one in pg 2 but the figure is on pg 5., so it make tedious to have to search the figure everytime It might be more useful to separate this figure into two Figure 1 (a, b) and figure 2 (c, d).

The entire Material and Method section was moved to the SI. Now, Figure 1 should be very close to page 2.

Figure S4 should be in the main manuscript. Moreover, in the text it is said "The highest intensity peak at 399.3 eV (A) is followed by 2 smaller features around 401 eV (B and C)" but in the figure it could be not appreciate this two features. It could be useful to include a zoom of this regions as an inset.

We thank the reviewer for this input. We made the changes accordingly.

Reviewer 3 Report

nanomaterials-2278794

Polymeric carbon nitrides for photoelectrochemical applications: ring opening induced degradation

Florentina Iuliana Maxim , Eugenia Tanasa , Bogdan Mitrea , Cornelia Diac , Tomáš Skála , Liviu Cristian Tanase , Catalin Ianasi , Adrian Ciocanea , Åžtefan ANTOHE , Eugeniu Vasile , Eugenia Fagadar-Cosma , Serban Nicolae Stamatin

The authors prepared Carbon nitrides from polycondensation of melamine. They characterized the materials and tested for the photoelectrochemical CO2 reduction followed by post-mortem analysis. post-mortem FT-IR confirmed structural degradation. Particularly, nitrogen local structure was distorted during photoelectrochemical degradation, with carbon atoms bonded to three nitrogen atoms degrading to two nitrogen atoms. Another significant finding was the presence of methylene terminals in post-mortem materials. The study concludes that polymeric carbon nitrides are susceptible to photoelectrochemical degradation via ring-opening.

The work fits to the journal and is in the scope of the SI.

The work is interesting and well written. My only comment is about the CO2 activity. The authors did not provide the activity profiles, but they focus their work on the differences before and after the tests. Are the materials real active for the CO2 electroreduction? Please comment. Also did the change in electrolyte affects the performance of the materials?

Some other minor points are

Please address the novelty of your work better in introduction

The authors did not refer to the O detected with XPS. Why?

Line 227 please change cc with mL

Line 325 KHCO3 please subscript the 3

Some spaces are missing especially before the references in text

Did the authors perform XPS measurements after CO2 reduction?

Line 337 maybe 0.6 μΑ?? Please check

Author Response

Are the materials real active for the CO2 electroreduction? Please comment.

Degradation of solid state electrodes has two sides: (1) changes in the surface chemistry of the degraded electrode and (2) releasing of fractions in the electrolyte. The manuscript at-hand is focused on the former, that is investigation of the solid state electrode after photoelectrochemical degradation. Initial product measurements were done by a GC coupled to a BDI and FID. The results were found to be inconsistent and were not deemed publishable without additional experiments. Such additional experiments include isotope labeling and post-mortem XPS. Therefore, we kindly rebut the comment at this stage as it is part of a future manuscript.

Also did the change in electrolyte affects the performance of the materials?

Both electrolytes have the same pH. We chose sodium sulphate as there was no literature to suggest that there is a difference in activity between the two electrolytes. It is challenging to predict if the material remains similar without further experimental evidence which falls beyond the general scope of this manuscript.

Some other minor points are

Please address the novelty of your work better in introduction

The changes were made and were marked yellow in the revised document.

The authors did not refer to the O detected with XPS. Why?

The signal of O 1s in XPS can have at least two sources: (1) adventitious carbon, and (2) oxygen linked to the samples. It is challenging to distinguish between the two types of oxygen. We decided to focus the results and discussion only on the signals that we were sure to arise from the samples in question. We kindly rebut this comment.

Line 227 please change cc with mL

The changes were made and were marked with yellow in the revised document.

Line 325 KHCO3 please subscript the 3

The changes were made and were marked yellow in the revised document.

Some spaces are missing especially before the references in text

Changes have been made and highlighted accordingly.

Did the authors perform XPS measurements after CO2 reduction?

Post-mortem XPS is part of a future work. Please see the first point of this answer. We kindly rebut the comment at this stage.

Line 337 maybe 0.6 μΑ?? Please check

The changes were made and were marked yellow in the revised document.